# Sex differences in saliva-based DNA methylation changes and environmental stressor in young African American adults

**Forough Saadatmand[1]ᵒ, Muneer Abbas[2]ᵒ, Victor Apprey[3], Krishma Tailor[3], Bernard Kwabi-Addo [3]***

**1** Department of Pediatrics, Howard University, Washington, D.C., United States of America, **2** Department of Microbiology & The National Human Genome Center, Howard University, Washington, D.C., United States of America, **3** Department of Biochemistry and Molecular Biology, College of Medicine, Howard University, Washington, D.C., United States of America

ᵒ These authors contributed equally to this work.
* Bkwabi-addo@howard.edu

## Abstract

### Background

Low socioeconomic status neighborhood exposure to stress and violence may be sources of negative stimuli that poses significant health risks for children, adolescents and throughout the life course of an individual. The study aims to investigate if aberrant epigenetic DNA methylation changes may be a potential mechanism for regulating neighborhood exposures and health outcomes.

### Methods

Exposure to environmental stressors identified in 98 young African American (AA) adults aged 18–25 years old from the Washington D.C., area were used in the study. We correlated the association between stress markers; cortisol, CRP, IgG, IGA, IgM, and self-reported exposure to violence and stress, with quantitative DNA methylation changes in a panel of gene-specific loci using saliva DNA.

### Results

In all participants studied, the exposure to violence was significant and negatively correlated with DNA methylation of *MST1R* loci (p = 0.032; r = -0.971) and nominally significant with *NR3C1* loci (p = 0.053; r = -0.948). In addition, we observed significant and negative correlation of DNA methylation changes of *LINE*1 (p = 0.044; r = -0.248); *NR3C*1 (p = 0.017; r = -0.186); *MSTR*1 (p = 0.022; r = -0.192); and *DRD*2 (p = 0.056; r = -0.184; albeit nominal significant correlation) with IgA expression. On the other hand, we observed a significant and position correlation of DNA methylation changes in *DRD*2 (p = 0.037; r = 0.184) with IgG expression. When participants were stratified by sex, we observed in AA young male adults, significant DNA methylation changes of MST1R (p< 0.05) and association with exposure to violence and IgG level. We also observed significant DNA methylation levels of DRD2 (p<

**Data Availability Statement:** All relevant data are within the paper and its Supporting Information files.

**Funding:** This study was supported by the following: 1. National Institute on Minority Health and Health Disparities of the National Institutes of Health under Award Number 2U54MD007597 2. The "Biological and Social Correlates of Drug Use in African American Adults" (BADU) dataset was supported by the National Institute on Minority Health and Health Disparities under grant #5P20MD000198 The funders had no role in study design, data collection and analysis, decision to publish, or preparation of the manuscript.

**Competing interests:** The authors have declared that no competing interests exist.

0.05) and association with IgA, IgG, and cortisol level. Furthermore, we observed significant DNA methylation changes of NR3C1 (p< 0.05) with stress, IgA, and IgG in the male participants only. On the other hand, we only observed significant and a positive association of IgG with DNA methylation levels of *ESR*1 (p = 0.041) in the young AA female participants.

## Conclusion

Our preliminary observation of significant DNA methylation changes in neuronal and immune genes in saliva samples supports our recently published genome-wide DNA methylations changes in blood samples from young AA male adults indicating that saliva offers a non-invasive means for DNA methylation prediction of exposure to environmental stressors in a gender-specific manner.

## Introduction

Low socioeconomic status (SES) neighborhoods may be a source of negative and stressful stimuli that pose significant health risks for children, adolescents, and the life course of individuals. Such low SES neighborhoods face a greater likelihood of exposure to ambient hazards. The exposure–disease paradigm has long suggested that differential "vulnerability" may modify the effects of toxicants on biological systems. There is physiological evidence to show that chronic exposure to neighborhood stressors including noise, violence, and poverty may disrupt endocrine signaling in the Hypothalamic Pituitary Adrenal axis (HPA) and the sympathetic nervous systems [1–3]. In the long term, these disruptions can contribute to the development of metabolic disorders [4] and chronic diseases Some studies suggest that families and individuals in low SES communities are exposed to unique chronic stressors that are highly correlated with cortisol expression levels [2, 5]. Furthermore, psychosocial stressors, such as exposure to interpersonal and community violence, may impact immune function [6]. Repeated episodes of acute or chronic psychological stress could induce an acute phase response and chronic inflammatory response(s) leading to many chronic diseases such as insulin resistance, atherosclerosis, hypertension, non-insulin-dependent diabetes mellitus type II and metabolic disorders [7]. For instance, racial and ethnic minorities that reside in low SES suffer disproportionately high rates of chronic disease morbidity and mortality from hypertension [8]. Furthermore, minority children and youth living in segregated neighborhoods tend to have higher rates of mortality, morbidity, and health risk factors compared with white neighborhoods, even after accounting for economic and other characteristics [9] and this underscores the lifetime health disparities among US minority populations. Exposure to acute stressors may result in higher cortisol, which may have negative metabolic health outcomes and conversely, exposure to chronic stressors may result in lower cortisol secretion which may increase inflammation and negative health risks. Psychosocial stress can lead to acute and chronic changes in the functioning of body systems (e.g., immune) consequently, leading to poor health outcomes. One concept suggests the long-term effects of chronic disease risk due to physical and social exposures during gestation, childhood, adolescence, young adulthood and later adult life (e.g., pollutants, allergens, and stress) [10]. Thus, what the growing fetus gets exposed to in utero environment can prime its own immune system toward asthma and atopy, contributing to related disorders in the offspring.

A cumulative risk model suggests that psychosocial experiences related to childhood poverty and associated adversities accumulating over a mother's life (e.g., family violence) might

have lasting effects on maternal psychological functioning [11] and stress responses [12], which, in turn, might affect the developing infant. While cortisol is central in maintaining homeostasis during acute stress [13], exposure to chronic stressors can result in dysregulation of the HPA axis through either hyper- or hypo-cortisolism [14–16] and this may increase the risks for hypertension, insulin resistance, neuronal damage, immune disorders, mental health disorders, and disrupt the ability to deliver cortisol to sites of inflammation within the body [14–17]. Perceived adverse neighborhood environment conditions as a type of chronic psycho-social stress have also been associated with obesity [18–20] and cardiovascular diseases [20]. In our previous studies, we have observed that young men and women exposed to violence experienced depression and sleep disturbance [21].

Stress is a response of the central nervous system (CNS) to environmental stimuli perceived as a threat to homeostasis. The stress response involves a complex network of mechanisms essential for survival, mediated by neurotransmitters, peptide hormones, and endocrine hormones from the enteric nervous system (ENS), a branch of the autonomic nervous system that among other functions affects the production of interleukins (ILs). These molecules in turn modulate the humoral and cellular components of the intestinal immune system. The ENS contains both vagal and spinal sensory neurons, which play an essential role in the transference of information from the CNS to ENS and vice versa [22]. Gut homeostasis results from neuroimmune modulation by anti- and pro-inflammatory ILs, neurotransmitters, and endocrine hormones, all of which influence the generation of intestinal secretory immunoglobulins (Igs). Immunoglobins in turn affects intestinal inflammation and permeability, which are essential factors in the functional integrity of the gut under stress conditions. Studies indicate that neurotransmitters such as *HTR7*, and upon release, results in the production of pro-inflammatory cytokines from immune cells, such as microglial cells, dendritic cells, and monocytes [23, 24]. Cytokines, such as interleukin-6 (IL-6), are involved in the production of a variety of blood biomarkers, including cortisol, the major stress hormone [25, 26]. Cortisol in turn, is involved in the regulation of several biological processes, such as cellular metabolism and immune function [25]. During stress response, cortisol concentration is increased resulting in a shift towards an inflammatory and humoral immune response [27].

The immune system depends on Igs which are composed of five classes that includes immunoglobulin A, G, and M (IgA, IgG, and IgM). Previous work by Hoffman et al. [28] has shown that serum immunoglobulins are part of the adaptive immune system whereby IgM provides a rapid immune response required for tissue homeostasis, whereas IgG and IgA are long-lasting high-affinity antibodies, and mainly provides mucosal immunity. Immunoglobulin measurements are used for the diagnosis and monitoring of various diseases, including primary immunodeficiencies and autoimmune diseases. Aging is associated with an increased ratio of memory to naive B-cells [29], which may lead to lower IgM and higher IgA and IgG levels in older compared to younger individuals [30]. Furthermore, previous population-based studies have demonstrated lower IgG [31, 32] and IgM [31], but higher IgA [31, 32] levels in men compared to women. Among others, body mass index (BMI) and lifestyle-related factors such as alcohol consumption and smoking may impact serum immunoglobulin levels as well [33, 34]. Older age and male sex are associated with higher IgA, but lower IgM, and urges investigation of age- and sex-specific reference ranges of immunoglobulins. Other determinants of the association of Igs and ethnicity, diet, lifestyle, and cardio-metabolic factors is reviewed by Khan et al. [35].

Growing evidence suggests that epigenetic changes are a key mechanism whereby environmental stressors interact with the genome leading to stable changes in DNA structure, gene expression, and behavior. A recent review by Giurgescu et al. [36], that investigated differential DNA methylation changes in inflammation-related candidates' genes, neighborhood

environmental exposures, and increased risk for cardiovascular diseases reported association of neighborhood socioeconomic status, social environment, and crime with either global or gene-specific DNA methylation and one study found a significant association of neighborhood socioeconomic disadvantage and social environment with DNA methylation in inflammation-related candidate genes. Another review by Park et al. [37], aimed at evaluating the relationship between stress-associated epigenetic changes and depression identified the following genes to be correlated with depression: *NRC*31, *SLCA*4, *BDNF*, *FKBP*5, *SKA*2, *OXTR*, *LINGO*3, *POU3F*1, and *ITGB*1. More specifically, epigenetic changes in glucocorticoid signaling (e.g., NR3C1, FKBP5), serotonergic signaling (e.g., SLC6A4), and neurotrophin (e.g., BDNF) genes appear to be the most promising therapeutic targets to explore for psychotropic treatments as epigenetic DNA methylation is reversible.

Recently, we have carried out genome-wide DNA methylation profiles in blood collected from African American young adult males living in the Washington D.C area and identified that exposure to environmental violence significantly correlated with differential DNA methylation changes in genes associated with the central nervous system and immune functions [38]. In the present study we wanted to examine the association between exposure to violence, stress-related biomarkers and gene-specific DNA methylation changes in saliva samples from young African American adults in Washington D.C. Our rational wasto investigate select genes that play roles as regulators of HPA axis and previously reported to undergo differential methylation in response to different environmental stressors namely, *FKBP*5, *NR3C*1 and *DRD*2 and sex steroidal hormone *ESR*1 as well as *MST1R* that is responsive to physiological stress and *LINE*1 gene which is a global methylation marker. We hypothesize that DNA methylation changes may be related to the expression of stress-related genes in response to environmental exposures and this may be identified in non-invasive biospecimen including saliva.

## Materials and methods

### Patient sample

A subset of patient population from a study entitled Biological and Social Correlates of Drug Use in African American Emerging Adults (BADU) was used in this study. The BADU study consisted of 557 African American young adults, ages 18–25 (Female = 274, Male = 283) who were recruited from socioeconomically disadvantaged Wards (7 and 8) of Washington, DC. This cross-sectional study was conducted between 2010 and 2012 to explore the effects of genetic markers for alcohol and depression, violence exposure, and drug use in young African Americans. A subset (Female = 50; Male = 48) of this study population was identified using t-distributed Stochastic Neighbor Embedding (tSNE) on the top 50 factors of the dataset, and the top 50 individuals with high Exposure to Violence (ETV) (Female = 25, Male = 25) and the bottom 48 individuals with no or exceptionally low ETV (Female = 25, Male = 23). Participants consented to and contributed blood and saliva samples for the studies. Approximately 2ml of the participant's saliva was deposited (spit) directly into an Oragene DNA (DNA Genotek Inc. Kanata ON, Canada) collection tube which contained DNA preservative. Consent was obtained from each participant in the study, and approval from Howard University's Institutional Review Board was obtained (Approval number: IRB-16-MED-03). Participants received $75 for their participation in this study.

### Bioassay analysis of BADU specimen

Serum levels of IgG, IgA, and IgM were measured in blood samples and cortisol levels was measured in saliva from participants. Briefly, IgM, IgG and IgA were quantified from eluted DBS samples by standard sandwich ELISA methodology (Human IgG, IgM, and IgA ELISA

Ready-SET-Go!® Kits, eBioscience, San Diego, CA, USA; Tecan GENios Pro Reader platform, Tecan, San Jose, CA, USA) as described by [39]. Saliva (25uL) was used to measure cortisol level using a commercially available, high sensitivity, competitive amino assay (Salimetrics, State College, PA) and according to the manufacturer's recommended protocol [40]. For accurate ELISA bioassays, assays were done in duplicates.

*Genomic DNA extraction from saliva*: DNA was manually isolated from saliva using the prepIT•L2P protocol, as per the manufacturer's instructions (DNA Genotek). Total DNA samples from saliva were quantified using the Nanodrop 1000 Spectrophotometer (Thermo Scientific, Waltham, MA).

## Bisulfite modification and DNA methylation analysis

Quantitative DNA methylation analysis- High molecular weight genomic DNA extracted from saliva was modified using sodium bisulfite treatment [41]. Briefly, genomic DNA (2 μg) was denatured in 0.3 mol/L NaOH at 37˚C for 15 minutes; sodium bisulfite and hydroquinone were added to final concentrations of 3.1 mol/L and 0.5 mmol/L, respectively. The reaction was incubated at 50˚C for 16 hours and desalted using Wizard DNA purification resin (Promega) according to the instruction of the manufacturer. Bisulfite modification was completed by DNA desulfonation in 0.3 mol/L NaOH at 37˚C for 15 minutes. Modified DNA was precipitated with ethanol, washed in 70% ethanol, dried, and dissolved in 50 μL of TE buffer. The PCR primers were designed to assay the methylation status of CpGs within 0.5 kb from the transcription start site. The CpG islands interrogated were previously described; *ESR*1 and *LINE*1 [42], *MST*1R [43], *DRD*2 [44] and newly designed CpG island assays for *NR3C*1 (chromosomal location chr5: 143,404,044–143,404,076): Forward primer sequence Biotin- 5′AATT TTTTAGGAAAAAGGGTGG-3′; Reverse primer 5′- AACCCCTTTCCAAATAACACACT TC-3′; Sequencing primer 5′- AACTCCCCAATAAATCTAAAAC-3′ and *FKBP*5 (chromosomal location chr6:35,558,486–35,558,567): Forward primer 5′- GGATTTGTAGTTGGGA TAATAATTTGG-3′; Reverse primer Biotin 5′-TCTTACCTCCAACACTACTACTAAAA-3′ and Sequencing primer 5′- GGAGTTATAGTGTAGGTTT-3′ based on bisulfite modified sequence information. The complete list of primer information for all genes studied and the pyrosequencing assay conditions are shown in S1 Table. The integrity of the PCR product was verified on 1.5% agarose gels with ethidium bromide staining. The PCR product was immobilized on streptavidin-Sepharose beads (Amersham), washed, and denatured, and the biotinylated strands were released into annealing buffer containing the sequencing primer. Pyrosequencing was done using the PSQ HS96 Gold SNP Reagents on a PSQ 96HS machine (Qiagen). Bisulfite-converted DNA from saliva of participants and blank reactions, with water substituted for DNA, served as a negative control, and bisulfite-converted *Sss*I methylase—treated saliva DNA served as a positive control. Each bisulfite PCR and pyrosequencing reaction were done in duplicate.

## Statistical analysis

The methylation index at each gene promoter and for each sample was calculated as the average value of mC/ (mC + C; mC is methylated cytosine and C is unmethylated cytosine) for all examined CpG sites in the gene and expressed as the percentage of methylation. Statistical significance was judged by either Fisher T-test, Mann-Whitney *t* test or Pearson correlation. The Mann-Whitney t-test was used to compare DNA methylation changes in participant cases. The Pearson correlation test was used to determine the correlation between DNA methylation changes versus cortisol/IgA/IgG/IgM/Stress/or Exposure to violence (ETV) in participants. The R software was used for the Mann-Whitney and Pearson correlation analysis. Data

analysis was done using SPSS for Windows (version 18.0, SPSS) for the logistic regression. Significance was set at $P < 0.05$.

## Results

The present study is to interrogate the role of DNA methylation changes in a subset of patient population from a study entitled Biological and Social Correlates of Drug Use in African American Emerging Adults (BADU; [21]). The BADU study also collected data on stress as well as markers of stress (e.g., cortisol) and the immune system (IgA and IgG).

   We assessed the promoter methylation status in saliva DNA samples extracted from 98 young African American adults (Female = 50, Male = 48; 18–25 years old). There is no statistical difference between the number (%) difference between females and males or age, studied in this study cohort (Demographic/clinical description and characteristics of participants is shown in S2 Table). We analyzed a total of 6 genes: *ESR*1, *DRD*2, *LINE*1, *MST*1R, that we have previously demonstrated to be differentially methylated in prostate and breast cancers [42–44]. We have included in this study 2 new methylated assays; *NR3C*1 and *FKBP*5 which have been reported to demonstrated differential methylation in response to stress and violence [45–47]. These genes were selected based on their potential role in immune signaling, sex steroidal signaling pathways and neurotransmission. All genes are representatives of a variety of cellular pathways involved in HPA. We used pyrosequencing assays to analyze the methylation status of these genes in the saliva DNA samples from the young African American adults consisting of 50 females and 48 males. For each gene investigated, the percentage (%) of methylation at the specific gene promoter locus was measured and compared between the female and male samples (Fig 1). Overall, we observed low % methylation for *ESR*1, *DRD*2, *NR3C*1 and intermediate % methylation levels for *LINE*1 and *MST*1R, whereas we observed high % methylation level for *FKBP*5. When the methylation levels were stratified by sex, five of the six genes (*ESR*1, *LINE*1, *MST*1R, *DRD*2, *NR3C*1) did not show any significant difference in % methylation level between the males and females whereas *FKBP*5 was significantly higher in the female samples compared to the male samples (p = 0.0158; data shown in S1 Fig). To explore the correlation of the saliva DNA methylation changes with immune function, we examined the methylation status with IgA and IgG biomarkers (Fig 2). We observed a significant and negative correlation between DNA methylation changes and IgA for the *LINE*1 repetitive element (p = 0.044; r = -0.248), *MST*1R (p = 0.022; r = -0.192), *NR3C*1 (p = 0.017; r = -0.186) and a nominally significant and negative correlation of *DRD*2 (p = 0.062; r = -0.184) DNA methylation changes with IgA level. On the other hand, we observed a significant and positive correlation for *DRD*2 (p = 0.037; r = 0.184), *NR3C*1 (p = 0.002; r = 0.291) and a nominally significant DNA methylation of *ESR*1 (p = 0.061; r = 0.188) and IgG. Next, we analyzed DNA methylation changes and self-reported exposure to violence (ETV) or neighborhood stress (Fig 3). We observed a significant and negative correlation of ETV for *MST*1R methylation (p = 0.032) and nominally significant *NR3C*1 methylation (p = 0.053) whereas we did not detect an association of DNA methylation changes for the other genes and exposure to violence. Out of the six genes that were analyzed, we only observed a nominally significant and negative correlation between *MST*1R (p = 0.057) with neighborhood stress. Finally, we were interested to ascertain gender differences in DNA methylation levels and correlation with IgA, IgG, Stress, ETV and saliva cortisol levels (Table 1). For males, we observed a significant and negative correlation of DNA methylation of *NR3C*1 with IgA (p = 0.031), IgG (p = 0.001), and stress (p = 0.043). We also observed a nominally significant and negative correlation of DNA methylation of *MST*1R with IgA (p = 0.061), and exposure to violence (p = 0.016), whereas we observed positive correlation of DNA methylation in *MST*1R with

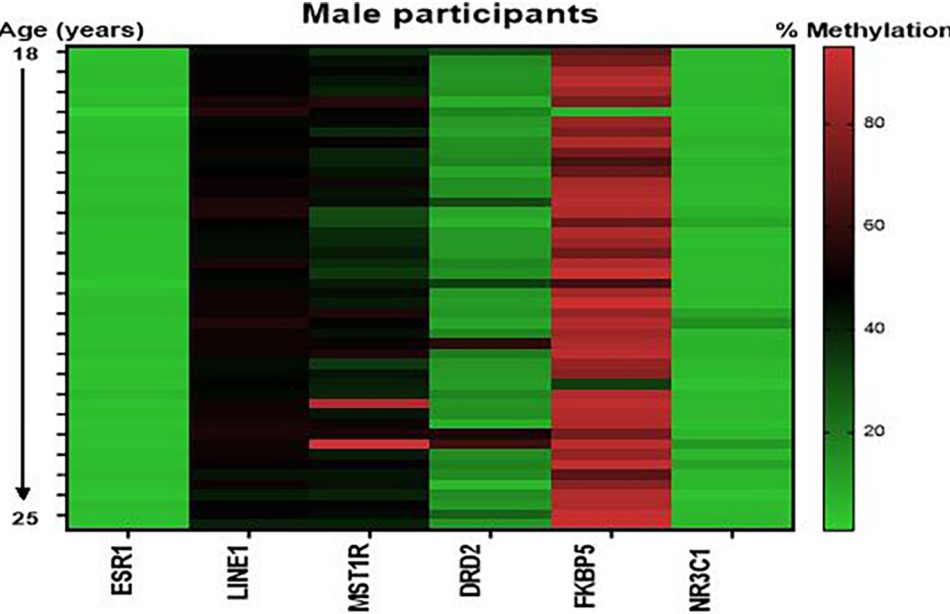

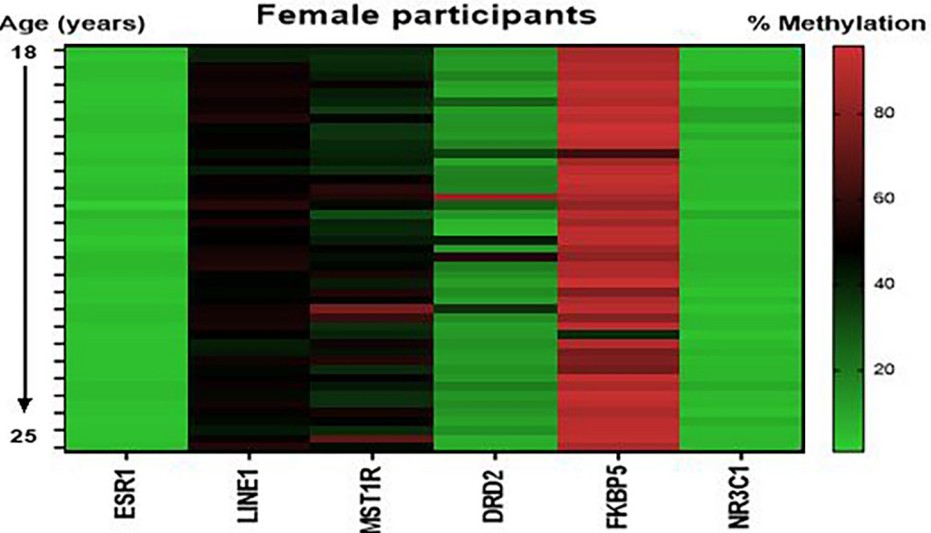

**Fig 1. A heat map of DNA methylation.** The percentage of DNA methylation levels at promoter CpG Island were analyzed in bisulfite-modified genomic DNA extracted from saliva from 98 African American young adults (Females = 50, Males = 48; Age 18–25 years old) using pyrosequencing. Top graph is samples from males and bottom is samples from females. Percentage (%) scale is shown to the right and samples are arranged by age from top to the bottom.

IgG (p = 0.038). Furthermore, in males, we observed significant and negative correlation of *DRD*2 methylation with IgA (p = 0.013) and positive correlation of *DRD*2 with IgG (p = 0.017) and saliva cortisol level (p = 0.029). For female samples, we only observed significant and positive correlation of DNA methylation for *ESR*1 and IgG (p = 0.041). Our data demonstrate differential and gender-specific DNA methylation changes in saliva and exposure to environmental stressors in the young AA adults studied.

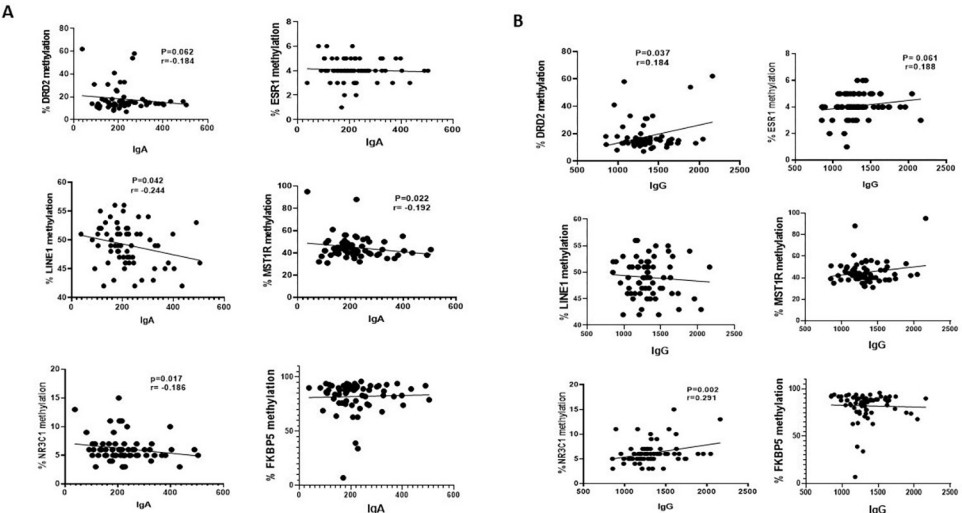

**Fig 2. Quantitative DNA methylation analysis and correlation with Immune markers.** The percentage of DNA methylation levels at promoter CpG island were compared with serum IgA (A) and IgG (B) levels. Y axis, percentage of methylated cytosines in the samples as obtained from pyrosequencing. X axis, IgA, or IgG serum levels. P value is indicated for each gene (Mann-Whitney T-test).

## Discussion

Early life stresses in humans (i.e. maltreatment, violence exposure, loss of a loved one) can trigger changes in epigenetic program of acquired behavioral traits via alterations in expressions of genes involved in the HPA axis that can even result in altered transgenerational transmission of altered behaviors [48]. This study sought to explore whether social/environmental stressors such as exposure to interpersonal and community violence as measured by immune, neurotransmitter, and stress biomarkers cause detectable changes in saliva DNA methylation

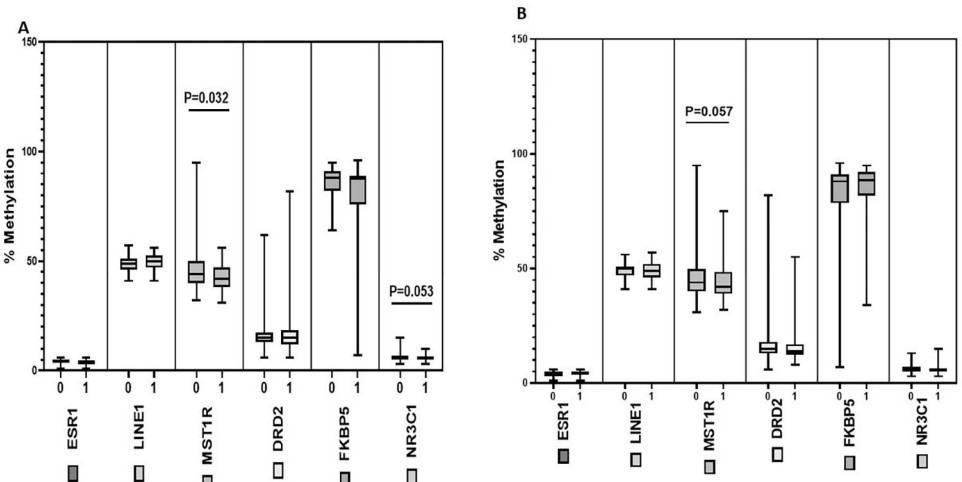

**Fig 3. Quantitative DNA methylation analysis and correlation with self-reported exposure to violence and stress.** The percentage of DNA methylation levels at promoter CpG island were compared with exposure to violence (A) and stress level (B). Y axis, percentage of methylated cytosines in the samples as obtained from pyrosequencing. X axis, exposure to violence (0 = no exposure; 1 = exposure to violence) or stress (0 = no stress, 1 = stress). P value is indicated for genes that shows significance and nominal significance (p <0.05; Mann-Whitney T-test).

**Table 1. The DNA methylation analysis of genes and correlation with biomarkers in both males and females are shown.**

| Male Dependent variable | Biomarker | Beta | Std. Error | Sig. | Female Dependent variable | Biomarker | Beta | Std. Error | Sig |
|---|---|---|---|---|---|---|---|---|---|
| NR3C1 | IgA | -.008 | .004 | **0.031** | NR3C1 | IgA | -.003 | 0.005 | .538 |
| | IgG | .005 | .001 | **0.001** | | IgG | .001 | .002 | .609 |
| | Cortisol | -.003 | .134 | .985 | | Cortisol | .124 | .129 | .347 |
| | ETV | -.863 | .693 | .219 | | ETV | -.029 | .535 | .958 |
| | Stress | -1.969 | .946 | **.043** | | Stress | -.505 | .713 | .871 |
| | | | | | | | | | |
| MST1R | IgA | -.040 | .021 | .061 | MST1R | IgA | -.018 | 0.018 | .326 |
| | IgG | .015 | .007 | **0.038** | | IgG | .001 | .008 | .924 |
| | Cortisol | .481 | .742 | .521 | | Cortisol | .379 | .441 | .400 |
| | ETV | -8.486 | 3.372 | **.016** | | ETV | -.587 | 2.622 | .824 |
| | Stress | -8.002 | 4.628 | .091 | | Stress | -6.132 | 3.448 | .083 |
| | | | | | | | | | |
| DRD2 | IgA | -.053 | .020 | **.013** | DRD2 | IgA | .004 | .019 | .821 |
| | IgG | .017 | .007 | **.017** | | IgG | -.010 | .008 | .262 |
| | Cortisol | 1.651 | .727 | **.029** | | Cortisol | -.605 | .618 | .341 |
| | ETV | -5.716 | 3.795 | .139 | | ETV | 1.238 | 4.399 | .780 |
| | Stress | -3.576 | 5.173 | .493 | | Stress | 2.580 | 5.544 | .575 |
| | | | | | | | | | |
| ESR1 | IgA | -.001 | .002 | .522 | ESR1 | IgA | -.002 | .002 | .342 |
| | IgG | .001 | .001 | .269 | | IgG | .002 | .001 | **0.041** |
| | Cortisol | .043 | .062 | .499 | | Cortisol | .068 | .044 | .138 |
| | ETV | -.022 | .290 | .940 | | ETV | .151 | .287 | .601 |
| | Stress | -.377 | .396 | .347 | | Stress | -.485 | .364 | .189 |
| | | | | | | | | | |
| LINE1 | IgA | -.018 | .006 | **.003** | LINE1 | IgA | .003 | .010 | .801 |
| | IgG | .000 | .002 | .846 | | IgG | .00007 | .004 | .985 |
| | Cortisol | .261 | .199 | .197 | | Cortisol | .177 | .244 | .476 |
| | ETV | -.519 | 1.103 | .640 | | ETV | 1.096 | 1.211 | .370 |
| | Stress | 1.286 | 1.507 | .398 | | Stress | .111 | 1.612 | .946 |
| | | | | | | | | | |
| FKBP5 | IgA | -.010 | .030 | .740 | FKBP5 | IgA | .022 | .031 | .483 |
| | IgG | .002 | .010 | .802 | | IgG | -.012 | .013 | .358 |
| | Cortisol | .997 | 1.060 | .353 | | Cortisol | -.786 | .749 | .306 |
| | ETV | -3.787 | 4.555 | .410 | | ETV | 2.923 | 2.802 | .303 |
| | Stress | -9.263 | 6.222 | .144 | | Stress | -1.618 | 3.730 | .667 |

Statistical significance (Sig.), Standard (Std.) errors and beta values are shown.

in young African American adults. In this study, we evaluated the promoter-specific DNA methylation of several genes that are key regulators of the HPA axis such as *FKBP*5 [49] and *NR3C*1, a glucocorticoid receptor gene, that plays a role in adaptation to stress [50]. The dopaminergic system, of which dopamine D2 receptor (*DRD*2) is closely associated with locomotion, reward, and memory [51] was analyzed as stress directly influences several fundamental behaviors and phenomena including locomotor activity, sexual activity, appetite, drugs or abuse via the dopaminergic system [52]. In addition, we investigated estrogen receptor 1 (*ESR*1) whose transcriptional activity has been identified as a potential candidate for the enduring and sex-specific effects of chronic adolescent stress exposure [53]. In addition, we

investigated the methylation status of *LINE*1, a repetitive element, a global methylation marker. It has been shown that LINE-1 transposition may play a role in differentiation of neurons during brain development as LINE-1 retrotransposition has been identified during adult neurogenesis in the human hippocampus and differential methylation patterns of *LINE*1 that might be involved in response to stress and resilience (Review by Misiak et al. [54]). Finally, we investigated DNA methylation status of a receptor tyrosine kinase, RON (*MST*1R) whose signaling is altered in response to physiological stress (e.g., serum starvation) [55]. Quantitative analysis of DNA methylation changes shows low methylation levels for ESR1 and NR3C1 and high methylation levels for LINE1, MST1R, DRD2 and FKBP5 in both the female and male participants. There was no age or sex-dependent methylation in five out of the six promoter methylation loci. Only promoter FKBP5 showed significantly higher methylation level in male participants compared to the female participants, suggesting that differential dysregulation of FKBP5 could alter immune response to stress in males and the female participants.

In this study, we observed a significant and a negative correlation between *LINE*1 and IgA levels only in the male participants suggesting that DNA methylation of *LINE*1 might play a role in stress response mediated by the biomarkers of the immune function, IgA specifically in males. The DNA methylation of *ESR*1 was significant and positively correlated with IgG in the female participants only, demonstrating that sex-specific effects of stress exposure via IgG are more pronounced in females than in males. The methylation of *DRD*2 showed a significant and negative correlation with IgA and significant positive correlation with IgG in only male participants. In addition, we observed significant positive correlation of *DRD*2 methylation with saliva cortisol expression levels only in the male participants demonstrating pronounced effects of *DRD*2 methylation in response to stressors in male participants. The DNA methylation of *MST*1R showed significant and a negative correlation with IgA in male participants only and could also be an indicator for physiological stressors. The DNA methylation level of *NR3C*1 showed a negative correlation with IgA and stress only in the male participants. On the other hand, we observed a positive correlation between *NR3C*1 DNA methylation and IgG levels only in the male participants demonstrating a role for *NR3C*1 methylation and adaptation to stress in the male participants. Finally, we observed that in males, DNA methylation of *FKBP*5 was significant and negatively correlated with IgA, IgG, and stress suggesting that FKBP5 methylation was predictor for stress markers through IgA, IgG, and exposure to stress. Overall, we observed that with the exception of *ESR*I methylation, which was significantly associated with IgG level in females, DNA methylation of the other five genes studied was not significantly associated with female sensitivity to environmental stressors, whereas the methylation of these genes was significantly associated with male sensitivity to environmental stressors. The data indicates a link between DNA methylation changes and immunoglobulins that may in part capture immune adaptation and exposure to environmental stressors.

Our observation of significant and differential DNA methylation in a panel of genes with biological roles in neuronal and immune signals using saliva samples from young AA male adults supports our recent genome-wide DNA methylation analysis [38] in blood samples from a subset of the young AA male adults which demonstrated differential methylated genes with biological functions enriched in central nervous and immune signaling pathways. The genome-wide analysis did not include female samples, so we are not able to make any comparison with our current observation. Thus, saliva offers a non-invasive means for DNA methylation prediction of exposure to environmental stressors.

Future work should evaluate changes in DNA methylation changes of other genes that may be more sensitive in female participants, perhaps genes that are important players in the estrogen signaling pathway. However, we cannot rule out the possibility that the young female cohort in our studies has buffers that modulate their stress levels which cannot be identified by

the markers studied here and this will require additional studies of environmental exposure and other socio-demographic, social, and psychosocial factors or other physiological markers.

## Limitations

This study is a preliminary observation and is a cross-sectional study, therefore it needs validation in a large cohort to confirm the associations with the determinants in our cohort.

## Supporting information

**S1 Fig. The percentage of DNA methylation levels at promoter CpG Island were analyzed in bisulfite-modified genomic DNA extracted from saliva from 98 African American young adults (Females = 50, Males = 48; Age 18–25 years old) using pyrosequencing.** Methylation levels stratified by sex shows significant difference between males and females for FKBP5 (p = 0.015; Fisher T-test).
(JPG)

**S1 Table. The complete list of primer information for all genes studied and the pyrosequencing assay conditions.**
(DOCX)

**S2 Table. Demographic and/or clinical description and characteristics of participants used in study.**
(DOCX)

## Acknowledgments

We thank the Howard University Hospital Clinical Research Unit for sample collection and the National Human Genome Center at Howard University for making these samples available for this work.

## Author Contributions

**Conceptualization:** Forough Saadatmand, Bernard Kwabi-Addo.

**Data curation:** Muneer Abbas, Bernard Kwabi-Addo.

**Formal analysis:** Victor Apprey, Bernard Kwabi-Addo.

**Funding acquisition:** Bernard Kwabi-Addo.

**Investigation:** Bernard Kwabi-Addo.

**Methodology:** Forough Saadatmand, Muneer Abbas, Krishma Tailor, Bernard Kwabi-Addo.

**Resources:** Forough Saadatmand.

**Writing – original draft:** Bernard Kwabi-Addo.

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
