## [Decision Letter · Decision Letter 0]

24 May 2022

PONE-D-22-11625Correlating Saliva-based DNA Methylation Changes and Environmental Stressor in Young African American AdultsPLOS ONE

Dear Dr. Kwabi-Addo,

Thank you for submitting your manuscript to PLOS ONE. After careful consideration, we feel that it has merit but does not fully meet PLOS ONE’s publication criteria as it currently stands. Therefore, we invite you to submit a revised version of the manuscript that addresses the points raised during the review process.

We look forward to receiving your revised manuscript.

Kind regards,

Saurabh Agarwal

Academic Editor

PLOS ONE

Journal Requirements:

When submitting your revision, we need you to address these additional requirements.1.

“This work was supported by the National Institute on Minority Health and Health Disparities of the National Institutes of Health under Award Number 2U54MD007597. The “Biological and Social Correlates of Drug Use in African American Adults” (BADU) dataset was supported by the National Institute on Minority Health and Health Disparities under grant #5P20MD000198. The content is solely the responsibility of the authors and does not necessarily represent the official views of the National Institutes of Health.”

“This study was supported by the following:

1. National Institute on Minority Health and Health Disparities of the National Institutes of Health under Award Number 2U54MD007597

2. The “Biological and Social Correlates of Drug Use in African American Adults” (BADU) dataset was supported by the National Institute on Minority Health and Health Disparities under grant #5P20MD000198

4. Please include your tables as part of your main manuscript and remove the individual files. Please note that supplementary tables (should remain/ be uploaded) as separate "supporting information" files

6. We noticed you have some minor occurrence of overlapping text with the following previous publication(s), which needs to be addressed:

-https://www.ncbi.nlm.nih.gov/pmc/articles/PMC3709436/

-https://www.frontiersin.org/articles/10.3389/fimmu.2021.664526/full

In your revision ensure you cite all your sources (including your own works), and quote or rephrase any duplicated text outside the methods section. Further consideration is dependent on these concerns being addressed.

Reviewers' comments:

Reviewer's Responses to Questions

**Comments to the Author**

1. Is the manuscript technically sound, and do the data support the conclusions?

Reviewer #1: Partly

Reviewer #2: Yes

2. Has the statistical analysis been performed appropriately and rigorously? 

Reviewer #1: I Don't Know

Reviewer #2: Yes

3. Have the authors made all data underlying the findings in their manuscript fully available?

Reviewer #1: Yes

Reviewer #2: Yes

4. Is the manuscript presented in an intelligible fashion and written in standard English?

Reviewer #1: No

Reviewer #2: Yes

5. Review Comments to the Author

Reviewer #1: This was a small but important investigation of DNA methylation of 5 candidate genes and LINE1 repeats, sex stratified, in relation to IgG, IgA, and cortisol levels as well as exposure to violence. The authors identify a significant inverse association with exposure to violence and MST1R methylation specifically in males. Other significant associations were observed between Ig levels or cortisol levels and one or more candidate genes, mostly in males. While these results are potentially novel and important to the field, a number of moderate to minor concerns should be address.

1. Despite sex differences in the effects of exposure to violence being profound in the results, the abstract does not describe the results of sex stratified analyses at all. The Results section of the abstract should be rewritten to better focus on the most significant findings with and without sex stratification. The significant association of MST1R methylation inversely associated with ETV in males would be one to especially highlight in the abstract for its novelty. The authors should also consider adding the finding of sex differences to the title.

2. Figure 1 legend mentions a p value by Fisher’s T-test, but there are no apparent p values or need for statistical testing from the figure, which is described as a heat map of DNA methylation levels at each gene locus.

3. In several places in the text, figures, and Table 1, p values >0.05 are described as significant. The authors can describe these as “nominally significant” or “trending” but the word “significant” should be reserved for p values <0.05. If the authors wish to highlight trends that approached significance in the figures and table, they should use a different designation/color from those that are <0.05.

4. There is some missing or confusing information in the Methods:

a. It is unclear from the methods when the blood was collected. Were blood and saliva collected from the original BADU cohort for antibody and cortisol measurements? If so, has that data been described previously for the larger cohort? Was there a new Oragene kit saliva collection for the 98 participants selected for DNA methylation, or was that done on archived saliva?

b. The PCR conditions should be given for the pyrosequencing assays, particularly the number of cycles.

c. A table providing the number of CpGs covered and their position relative to the transcription start site of each gene in the pyrosequencing analyses would be helpful. Including the full gene name and/or function in this table would also be helpful.

5. References should be provided to support this sentence: “We have included in this study 2 new methylated assays; NR3C1 and FKBP5 which have been reported to demonstrated differential methylation in response to stress and violence.”

6. Throughout the text, consider replacing “direct correlation” with “positive correlation” for clarity.

7. The manuscript should be edited for grammar and punctuation.

8. In Table 1, the label “cortisol” should replace “stress” to be more specific about what was measured.

Reviewer #2: -reference 8 is a study om internet use and does not appear to be an appropriate reference to substantiate the statement of line 66-67

-reference 9 is almost 20 years old. is there a more current epidemiological study that could be cited?

-given your previous work in this population with DNAm, it would be good to compare and contrast in the discussion the findings here versus those in the EWAS.

-Were the bioassay ELISAs performed in duplicate to estimate CV%?

-thank you for providing specifics on the biotin, sequencing primer and choromosomal locations for primer sets. Can you provide the Sequence to analyze for the PSQ assay so that others can replicate?

-can you provide the mean with SD for each CpG site as well as your regional averaged CpG methylation for each gene and LINE-1? This will help readers understand the levels of methylation in your measurement across the CpG collectively and individually.

-as this is a candidate gene study, it should be relatively easy to understand the rationale for your gene selection. This is in the introduction to a certain extent but the introduction does name many more genes you did not look at. How did you come to your final list of candidate genes for analysis?

-although this study utilizes a previously existing set of samples for analysis, can you provide a brief table or narrative of the major demographics/clinical characteristics for this sample set?

-should there be a consideration of false positives with so many statistical tests? I don't necessarily think you have to do bonferroni correction but a consideration would be good in the discussion.

-limitations should also state the cross-sectional nature, lack of control group and potential for other genes or other areas within the selected genes to show associations not found.

6. PLOS authors have the option to publish the peer review history of their article (what does this mean?). If published, this will include your full peer review and any attached files.

Reviewer #1: **Yes: **Janine LaSalle

Reviewer #2: No

---

## [Author Response · Author response to Decision Letter 0]

7 Jul 2022

Here is our response to comments raised by reviewer 1:

1. Despite sex differences in the effects of exposure to violence being profound in the results, the abstract does not describe the results of sex stratified analyses at all. The Results section of the abstract should be rewritten to better focus on the most significant findings with and without sex stratification. The significant association of MST1R methylation inversely associated with ETV in males would be one to especially highlight in the abstract for its novelty. The authors should also consider adding the finding of sex differences to the title.

Response: we have revised our title; the new title is “Sex Differences in Saliva-based DNA Methylation Changes and Environmental Stressor in Young African American Adults”. We have re-written the results section in the abstract to emphasize the significant differences in the correlation between environmental stressors and DNA methylation changes in participants stratified and non-stratified by sex.

2. Figure 1 legend mentions a p value by Fisher’s T-test, but there are no apparent p values or need for statistical testing from the figure, which is described as a heat map of DNA methylation levels at each gene locus.

Response: We include supplementary figure S1 that shows significant DNA methylation changes for FKBP5 when stratified by sex and the p value.

3. In several places in the text, figures, and Table 1, p values >0.05 are described as significant. The authors can describe these as “nominally significant” or “trending” but the word “significant” should be reserved for p values <0.05. If the authors wish to highlight trends that approached significance in the figures and table, they should use a different designation/color from those that are <0.05.

Response: This has been corrected to show in places where the p values are > 0.05 as nominally significant.

4. There is some missing or confusing information in the Methods:

a. It is unclear from the methods when the blood was collected. Were blood and saliva collected from the original BADU cohort for antibody and cortisol measurements? If so, has that data been described previously for the larger cohort? Was there a new Oragene kit saliva collection for the 98 participants selected for DNA methylation, or was that done on archived saliva? b. The PCR conditions should be given for the pyrosequencing assays, particularly the number of cycles. c. A table providing the number of CpGs covered and their position relative to the transcription start site of each gene in the pyrosequencing analyses would be helpful. Including the full gene name and/or function in this table would also be helpful.

Response: We indicate in the materials and methods that blood and saliva were collected from the participants at the same time. We have removed confusion about the Oragene kit as the same oragene kit was used on all samples. We indicate in the material and method section that the pyrosequencing assay conditions are provided in supplementary Table S1. 

5. References should be provided to support this sentence: “We have included in this study 3 new methylated assays; NR3C1 and FKBP5 which have been reported to demonstrated differential methylation in response to stress and violence.” 

Response: We have provided references to support this statement (line 247).

6. Throughout the text, consider replacing “direct correlation” with “positive correlation” for clarity.

Response: We have replaced direct and inverse correlations to positive and negative correlations respectively

7. The manuscript should be edited for grammar and punctuation.

Response: We have been careful to edit this revised manuscript.

8. In Table 1, the label “cortisol” should replace “stress” to be more specific about what was measured.

Response: In this study we measured stress as self-reported experience such as exposure to interpersonal and community violence and stress markers (e.g., cortisol) as two separate entities as reported in Table 1 and figure 3.

Here is our response to comments raised by reviewer 2:

• reference 8 is a study om internet use and does not appear to be an appropriate reference to substantiate the statement of line 66-67

Response: we have corrected this erroneous reference with the appropriate reference (line 71)

• reference 9 is almost 20 years old. is there a more current epidemiological study that could be cited?

Response: this is corrected and replaced with a 2017 reference (line 73).

• given your previous work in this population with DNAm, it would be good to compare and contrast in the discussion the findings here versus those in the EWAS.

Response: We have included a paragraph in the discussion that our observation of significant and differential DNA methylation in a panel of genes with biological roles in neuronal and immune signals using saliva samples from young AA male adults supports our recent genome-wide DNA methylation analysis in blood samples from a subset of the young AA male adults. Thus, saliva offers a non-invasive means for DNA methylation prediction of exposures to environmental stressors. 

• Were the bioassay ELISAs performed in duplicate to estimate CV%?

Response: we indicate in the materials and methods section that for accurate ELISA bioassays, assays were done in duplicates. 

• thank you for providing specifics on the biotin, sequencing primer and choromosomal locations for primer sets. Can you provide the Sequence to analyze for the PSQ assay so that others can replicate?

Response: Both reviewer 1 and this reviewer made the same comment and we now provide the the primer information and pyrosequencing assay condition for all genes studied in supplementary table S1.

• can you provide the mean with SD for each CpG site as well as your regional averaged CpG methylation for each gene and LINE-1? This will help readers understand the levels of methylation in your measurement across the CpG collectively and individually.

Response: Pyrosequencing is a robust sequencing analysis that quantify CpG (as Cytocine/thymine polymorphism) and computed as the average % DNA methylation based on the number of CpG sites analyzed. 

• as this is a candidate gene study, it should be relatively easy to understand the rationale for your gene selection. This is in the introduction to a certain extent but the introduction does name many more genes you did not look at. How did you come to your final list of candidate genes for analysis?

Response: We emphasize our rational to investigate selected genes that play roles as regulators of HPA axis and previously reported to undergo differential methylation in response to different environmental stressors namely, FKBP5, NR3C1 and DRD2 and sex steroidal hormone ESR1 as well as MST1R that is responsive to physiological stress and LINE1 gene which is a global methylation marker in the introduction section (line 153-157) and also in the discussion. 

• although this study utilizes a previously existing set of samples for analysis, can you provide a brief table or narrative of the major demographics/clinical characteristics for this sample set?

Response: we provide descriptive characteristics of participants demographics and socioeconomics status as supplementary Table S2 (line 242-243).

• limitations should also state the cross-sectional nature, lack of control group and potential for other genes or other areas within the selected genes to show associations not found.

Response: we include the cross-sectional nature of our study as a limitation and therefore study needs validation in a large cohort to confirm the associations with the determinants in our cohort.

---

## [Decision Letter · Decision Letter 1]

12 Aug 2022

Sex Differences in Saliva-based DNA Methylation Changes and Environmental Stressor in Young African American Adults

PONE-D-22-11625R1

Dear Dr. Kwabi-Addo,

We’re pleased to inform you that your manuscript has been judged scientifically suitable for publication and will be formally accepted for publication once it meets all outstanding technical requirements.

Kind regards,

Saurabh Agarwal

Academic Editor

PLOS ONE

Additional Editor Comments (optional):

Reviewers' comments:

Reviewer's Responses to Questions

**Comments to the Author**

1. If the authors have adequately addressed your comments raised in a previous round of review and you feel that this manuscript is now acceptable for publication, you may indicate that here to bypass the “Comments to the Author” section, enter your conflict of interest statement in the “Confidential to Editor” section, and submit your "Accept" recommendation.

Reviewer #2: All comments have been addressed

2. Is the manuscript technically sound, and do the data support the conclusions?

Reviewer #2: Yes

3. Has the statistical analysis been performed appropriately and rigorously? 

Reviewer #2: Yes

4. Have the authors made all data underlying the findings in their manuscript fully available?

Reviewer #2: Yes

5. Is the manuscript presented in an intelligible fashion and written in standard English?

Reviewer #2: Yes

6. Review Comments to the Author

Reviewer #2: The authors have done a satisfactory job answering critiques and revising their paper. I do not have any further comments to provide.

7. PLOS authors have the option to publish the peer review history of their article (what does this mean?). If published, this will include your full peer review and any attached files.

Reviewer #2: No

---

## [Editor Report · Acceptance letter]

26 Aug 2022

PONE-D-22-11625R1 

Sex Differences in Saliva-based DNA Methylation Changes and Environmental Stressor in Young African American Adults. 

Dear Dr. Kwabi-addo:

I'm pleased to inform you that your manuscript has been deemed suitable for publication in PLOS ONE. Congratulations! Your manuscript is now with our production department. 

Kind regards, 

on behalf of

Dr. Saurabh Agarwal 

Academic Editor

PLOS ONE